# Typing of the Gut Microbiota Community in Japanese Subjects

**DOI:** 10.3390/microorganisms10030664

**Published:** 2022-03-20

**Authors:** Tomohisa Takagi, Ryo Inoue, Akira Oshima, Hiroshi Sakazume, Kenta Ogawa, Tomo Tominaga, Yoichi Mihara, Takeshi Sugaya, Katsura Mizushima, Kazuhiko Uchiyama, Yoshito Itoh, Yuji Naito

**Affiliations:** 1Molecular Gastroenterology and Hepatology, Graduate School of Medical Science, Kyoto Prefectural University of Medicine, Kyoto 602-8566, Japan; t-sugaya@koto.kpu-m.ac.jp (T.S.); k-uchi@koto.kpu-m.ac.jp (K.U.); yitoh@koto.kpu-m.ac.jp (Y.I.); 2Department for Medical Innovation and Translational Medical Science, Graduate School of Medical Science, Kyoto Prefectural University of Medicine, Kyoto 602-8566, Japan; 3Laboratory of Animal Science, Setsunan University, Osaka 573-0101, Japan; ryo.inoue@setsunan.ac.jp; 4Department of Research and Development, PreMedica Inc., Tokyo 105-0011, Japan; akira.oshima@nk-m.co.jp (A.O.); hiroshi.sakazume@premedica.co.jp (H.S.); kenta.ogawa@premedica.co.jp (K.O.); tomo.tominaga@premedica.co.jp (T.T.); y.mihara@premedica.co.jp (Y.M.); 5Department of Human Immunology and Nutrition Science, Graduate School of Medical Science, Kyoto Prefectural University of Medicine, Kyoto 602-8566, Japan; mizusima@koto.kpu-m.ac.jp (K.M.); ynaito@koto.kpu-m.ac.jp (Y.N.)

**Keywords:** gut microbiota community, partitioning around medoids (PAM) model, Dirichlet multinominal mixtures (DMM) model, *Bifidobacterium*, enterotype

## Abstract

Gut microbiota are involved in both host health and disease and can be stratified based on bacteriological composition. However, gut microbiota clustering data are limited for Asians. In this study, fecal microbiota of 1803 Japanese subjects, including 283 healthy individuals, were analyzed by 16S rRNA sequencing and clustered using two models. The association of various diseases with each community type was also assessed. Five and fifteen communities were identified using partitioning around medoids (PAM) and the Dirichlet multinominal mixtures model, respectively. Bacteria exhibiting characteristically high abundance among the PAM-identified types were of the family *Ruminococcaceae* (Type A) and genera *Bacteroides*, *Blautia*, and *Faecalibacterium* (Type B); *Bacteroides*, *Fusobacterium*, and *Proteus* (Type C); and *Bifidobacterium* (Type D), and *Prevotella* (Type E). The most noteworthy community found in the Japanese subjects was the *Bifidobacterium*-rich community. The odds ratio based on type E, which had the largest population of healthy subjects, revealed that other types (especially types A, C, and D) were highly associated with various diseases, including inflammatory bowel disease, functional gastrointestinal disorder, and lifestyle-related diseases. Gut microbiota community typing reproducibly identified organisms that may represent enterotypes peculiar to Japanese individuals and that are partly different from those of indivuals from Western countries.

## 1. Introduction

Recently accumulated information regarding human gut microbiota has revealed that their composition is linked to host health and various diseases, including inflammatory bowel disease (IBD), irritable bowel syndrome (IBS), and allergies, as well as lifestyle-related diseases, such as obesity and nonalcoholic fatty liver disease (NAFLD) [1,2,3,4]. Arumugam et al. demonstrated that human gut microbiota can be classified into three clusters (enterotypes) based on their bacteriological composition using partitioning around medoids (PAM) [5]. Three traditional enterotypes are characterized by high levels of *Bacteroides* (enterotype 1), *Prevotella* (enterotype 2), and *Ruminococcus* (enterotype 3). These enterotypes have been shown to be independent of age, gender, cultural background, and geography but are associated with long-term diet. For instance, *Prevotella* is strongly associated with a carbohydrate-rich diet, whereas the *Bacteroides* enterotype is associated with consumption of protein and animal fat, such as that common in a Western diet [6].

Community-type clustering of stool samples has also been proposed [7,8]. These reports describe the use of a Dirichlet multinomial mixtures (DMM) modeling approach to stratify stool into four clustered types: *Ruminococcaceae* (R), *Prevotella* (P), *Bacteroides 1* (B1), and *Bacteroides 2* (B2) enterotypes. The R enterotype is prevalent in hard stools, whereas the P enterotype is associated with loose stool [9]. The B1 enterotype is the most common enterotype in healthy populations adopting a Western diet, whereas the B2 enterotype is characterized by a high proportion of *Bacteroides* and a low proportion of *Faecalibacterium* and is more prevalent with systemic inflammation levels and IBD [10,11]. This suggests that the B2 enterotype may be indicative of an unhealthy microbiome constellation. Clustering of the human gut microbiota using Western cohorts seems to be useful in the assessment of their association with various diseases and to evaluate alterations in gut microbiota resulting from therapeutic intervention. However, it is also well known that the profile and structure of gut microbiota differ among residents of a particular region and based on ethnic differences. Nishijima et al. [12] confirmed that the gut microbiome of a Japanese population differs considerably from that of other populations. Accordingly, the stratification of gut microbiota on the dataset used by Nishijima et al. [12] revealed that the distribution of enterotypes was also characteristic in Japanese population compared to that of other countries [13]. This difference cannot be explained by diet alone. Meanwhile, an analysis of gut microbiota from school-age children in Asian regions revealed two enterotype-like clusters that were identified by variations of *Prevotella* or *Bifidobacterium/Bacteroides* [14]. Therefore, the gut microbiota structure is highly reflective of the country and region of residence of individuals, as well as their dietary habits and lifestyles. However, few studies regarding the compositional microbial profiles of Japanese populations have been conducted.

In the present study, we investigated the gut microbiota profiles of Japanese subjects. To the best of our knowledge, this is the first report regarding gut microbiota profiles based on a large number of Japanese subjects, including both healthy individuals and those with various diseases.

## 2. Materials and Methods

### 2.1. Study Subjects and Data Collection

A total of 1803 individuals of varying disease status were selected from our outpatient clinic from November 2016 to April 2017. The eligible subjects included both male and female subjects 14 years of age or older but younger than 101 years of age. The distribution of the 1803 enrolled study subjects, including 283 healthy individuals, is summarized in Table 1. The healthy subjects were considered to be in good health. The exclusion criteria were as follows: administration of antibiotics, corticosteroids, immunosuppressants, or acid-suppressing agents (proton pump inhibitors (PPIs) or histamine-type 2 receptor blockers (H2 blockers)) within the 3 months prior to collection of fecal samples or a history of underlying malignant disease. In addition, we excluded patients with serious metabolic, respiratory, cardiologic, renal, hepatic, hematologic, neurologic, or psychiatric functions, such and those who regularly used medications affecting intestinal motility, such as laxatives, antidepressants, opioid narcotic analgesics, anticholinergics, and prebiotic or probiotics. Individuals who were pregnant or lactating were also excluded. Patients with other factors, as evaluated by researchers, that could affect intestinal motility or gut microbiota were also excluded.

Hypertension (HT) was defined as systolic blood pressure ≥140 mmHg, diastolic blood pressure ≥90  mmHg, or the current use of antihypertensive medication. Hyperlipidemia (HL) was defined as a serum low-density lipoprotein cholesterol concentration ≥140 mg/dL, high-density lipoprotein cholesterol concentration < 40 mg/dL, triglyceride concentration ≥150 mg/dL, or the current use of cholesterol-lowering medication. Type 2 diabetes mellitus (T2D) was defined as a fasting plasma glucose level ≥126 mg/dL, hemoglobin A1c level ≥6.5%, or the current use of diabetes medication. Obesity was defined as a body mass index (BMI) ≥ 30 kg/m^2^.

### 2.2. Sample Collection and DNA Extraction

Fecal samples were collected, and gut bacterial composition analysis was performed according to previous reports [15,16,17]. Briefly, fecal samples the size of a grain of rice were collected using guanidine thiocyanate solution available in a feces collection kit (Techno Suruga Lab, Shizuoka, Japan). After vigorous mixing, the samples were stored at room temperature for a maximum of 7 days until DNA extraction.

Genomic DNA was isolated using a NucleoSpin microbial DNA kit (Macherey-Nagel, Düren, Germany). Approximately 500 µL of the stored fecal sample was placed in a microcentrifuge tube containing 100 µL of Elution Buffer BE. The mixture was then placed into a NucleoSpin beads tube with proteinase K and subjected to homgenization with mechanical beads for 12 min at 30 Hz in a TissueLyzer LT small bead mill. The subsequent extraction procedure was performed according to the manufacturer’s instructions. The extracted DNA samples were purified using an Agencourt AMPure XP system (Beckman Coulter, Brea, CA, USA).

### 2.3. Sequencing of the 16S rRNA Gene

Two-step polymerase chain reaction (PCR) was performed to generate sequence libraries of the purified DNA samples. The first PCR was performed to amplify the DNA samples using a 16S (V3–V4) metagenomic library construction kit for NGS (Takara Bio Inc, Kusatsu, Japan). The primer pair for the first PCR included the 341F forward primer (5’-TCGTCGGCAG CGTCAGATGT GTATAAGAGA CAGCCTACGG GNGGCWGCAG-3’) and 806R reverse primer (5’-GTCTCGTGGG CTCGGAGATG TGTATAAGAG ACAGGGACTA CHVGGGTWTC TAAT-3’) that corresponded to the V3-V4 region of the 16S rRNA gene. The second PCR was performed using a Nextera XT index kit (Illumina, San Diego, CA, USA) to add the index sequences for the Illumina sequencer with a barcode sequence. The prepared libraries were then subjected to sequencing of 250 paired-end bases at the Biomedical Center at Takara Bio using a MiSeq Reagent v3 kit and the MiSeq system (Illumina).

### 2.4. Microbiome Analysis and Community Typing

The obtained sequence data were processed using the standard QIIME2 (ver 2020.8) software pipeline. Denoising was performed using the DADA2 plugin to generate amplicon sequence variants (ASVs). Taxonomy assignment of each ASV was then conducted by a scikit-learn naïve Bayes machine-learning classifier trained on the Greengenes (13_8) 99% operational taxonomic units (OTUs). Singletons and ASVs assigned to chloroplast and mitochondria were removed for this study. The α-diversity indices, Chao1 index (ASV richness estimation), and Shannon index (ASV evenness estimation) were calculated using QIIME2. β-diversity at bacterial genus level was estimated using the Bray–Curtis metric to calculate distances between the samples and visualized using principal coordinate analysis (PCoA).

Two general approaches were used to assign the samples to community types1; the partitioning around medoids (PAM) model and the Dirichlet multinominal mixtures (DMM) model. Data at the genus level were used for community typing, and only frequently detected genera that were detected in more than 50% of the samples were selected. The genera not selected and considered infrequently detected genera were classified together as “other genera”. Similarly, all unclassified genera were grouped as “unclassified genera.” The optimal number of components (communities) was identified by selecting the number that gave the highest average silhouette width in the PAM-based clustering model and the minimum Laplace approximation to the negative log model in the DMM-model-based clustering. The maximum number of components evaluated was set at 20 in both clustering methods. Community typing was performed using R software (ver. 4.0.4) with the proper packages, such as “cluster” and “DirichletMultinomial.”

A model was constructed to predict each of the five PAM-identified communities based on composition of the frequently detected genera, plus “other genera” and “unclassified genera” using support vector machines (SVM) in JMP Pro (SAS Institute, Tokyo, Japan). The power of the prediction model was evaluated using holdback validation of approximately 65% of the training data (1200 subjects) and 35% of the prediction data (603 subjects). A linear kernel and cost parameter equal to 1 was applied in this study.

### 2.5. Statistical Analysis

The α-diversity indices among components were statistically compared using one-way analysis of variance (ANOVA) with a Tukey HSD post hoc test. Differences in β-diversity among components were evaluated using permutational analysis of variance (PERMANOVA), followed by a Tukey HSD post hoc test. The statistical analyses were performed using R software and the vegan package for R. Statistical differences (*p* < 0.05) in the relative abundance of bacterial phyla and genera among groups were evaluated using one-way ANOVA with the Benjamini–Hochberg correlation in STAMP software. The odds ratio of each disease for each of the gut microbiota was calculated based on the disease frequency in each community. Statistical differences (*p* < 0.05) in the odds ratio of the diseases in each community were calculated using the Wald test in JMP Pro.

### 2.6. Ethics Statements

The study conformed to the code of ethics stated in the Declaration of Helsinki, and the research protocol was approved by the Ethics Committee of Kyoto Prefectural University of Medicine (ERB-C-1770-2). All participants provided written informed consent prior to enrollment. The study was registered at the University Hospital Medical Information Network Center (UMIN000045216).

## 3. Results

### 3.1. Enrolled Study Participants

We enrolled 1803 Japanese subjects (983 male, 820 female), including 283 healthy subjects (Table 1). The average age of all participants was 64.2 ± 15.0 years (14–101 years). As shown in Table 1, the participants exhibited various diseases statuses, including cardiovascular diseases; hepatic diseases; functional gastrointestinal disorders; endocrine diseases; neurological diseases; psychiatric diseases; IBD; autoimmune diseases; malignant diseases; and lifestyle-related diseases, such as hypertension, dyslipidemia, hyperuricemia, T2D, and obesity. A more detailed breakdown of each disease is shown in Appendix A.

### 3.2. Gut Microbiota of Japanese Participants

Based on average relative abundance, the gut microbiota of Japanese subjects enrolled in the study predominantly consisted of four phyla: Firmicutes, Bacteroidetes, Actinobacteria, and Proteobacteria (Figure 1a), which is consistent with reports from previous studies [12,17]. At the genus level, seven genera were found to be predominant in the Japanese subjects, *Bacteroides, Bifidobacterium, Faecalibacterium, Blautia, Ruminoocccus* (family *Ruminococcaceae*), *Roseburia*, and *Prevotella*. The sum of their average abundance accounted for approximately 45% of the gut microbiota (Figure 1b). Thirty-six genera were found in >50% of the subjects (Figure 1c). These frequently detected genera were used for community typing of the gut microbiota.

### 3.3. Gut Microbiota Community Typing in Japanese Participants Enrolled in the Study

The optimal number of communities defined based on PAM clustering was five (Appendix A), whereas that based on DMM-model clustering was 15 (Appendix A). Characteristics of the 15 communities identified by the DMM model and their relations with the various diseases are briefly discussed below. However, for simplicity, the relationships between the five communities of the gut microbiota identified by the PAM model and the relevant diseases are mainly discussed in the subsequent portions of this article.

The five communities were categorized as type A–E in accordance with the order identified by the PAM clustering. As shown Table 2, the number of healthy individuals in the type E community was the largest among the five clusters, whereas the number of healthy subjects in the type A and type D communities was notably smaller compared with those in the other three types. The α-diversity indices (both Chao1 and Shannon indices) were the highest in type A, followed by those in type E and type C (Figure 2a,b). Both Chao1 and Shannon indices were lowest in the type D community. With respect to β-diversity, based on Bray–Curtis dissimilarity, the distance between type A and type B, that between type B and type E, and that between type C and type D were not significantly different (Figure 2c,d). However, the distances between other pairs, including between type A and type D, were significantly different.

The taxonomic characteristics of type A at the genus level were in highest abundance for *Coprococcus*, *Gemminger*, and *Roseburia* (Figure 3a, Appendix A). The abundance of *Ruminococcus* (family *Ruminococcaceae*) in type A was the second highest, next to that in type C. In addition, type A had the highest abundance of unclassified genera. Further evaluation revealed that the abundance of unclassified genera belonging to family *Ruminococcaceae* was highest in type A among the five types, with a mean abundance of 4.6% (Appendix A). The abundance of *Bacteroides* was highest in type B, whch also had the highest abundance of *Blautia* and *Faecalibacterium*. The second highest abundance of *Bacteroides* was in type C. In regards to “other genera,” type C exhibited the highest abundance of *Megamonus*, with a mean abundance of 4.9%. Type C also had the highest abundance of *Fusobacterium* and *Proteus*. The highest abundance of major lactic-acid-producing bacteria was in type D. The abundance of *Bifidobacterium*, *Lactobacillus*, and *Streptococcus* was significantly higher in type D compared with that in the other four types. A clear characteristic of type E was the highest abundance of *Prevotella*. The characteristic components of the gut microbiota in each type are summarized in Table 3. The SVM-based model generated using data from 1200 randomly selected study subjects as training data showed a highly accurate classification of the PAM-identified communities using prediction data from the remaining 603 study subjects. The rate of misclassification of the validation dataset was approximately 7.8% (47/603). The area under the curve (AUC) values of the receiver operating characteristic (ROC) curve for each type was >0.98 (Figure 4).

### 3.4. Association between Gut Microbiota Community Types and Disease Status

The odds ratio of each disease status was subsequently calculated based on the profile of type E (Figure 5). Type E was chosen because it had the largest population of healthy subjects among the five types, at 35.8% (Table 2). Type B, which included the second largest number of healthy individuals, at 26.6%, showed no significant difference in odd ratios of various diseases, except for IBD. In contrast, type A, type C, and type D, which included small numbers of healthy subjects, demonstrated associations with various diseases. Type A showed especially high odds ratios for cardiovascular diseases, neurological diseases, and lifestyle-related diseases, whereas type D showed high odds ratios for IBD and functional gastrointestinal disorders. A relatively high odds ratio was also revealed in type C for IBD; however, this was less that that in type D. Therefore, each type was found to be associated with various disease risks. More detailed odds ratio findings for each disease are shown in Appendix A.

### 3.5. Gut Microbiota Communities Identified via DMM-Based Clustering

Similarly the PAM-identified communities, each of the 15 communities identified through clustering based on the DMM model were sequentially labeled as type 1–15. The community type with the largest population of healthy subjects was type 10, with 42.2% healthy subjects (Appendix A). The characteristic bacterial taxonomy of type 10 includes the genus *Prevotella* in the highest abundance, which is consistent with type E of the PAM-identified communities (Appendix A). Calculation of the odds ratios of various diseases based on the profiles of type 10 revealed types 5, 11, 12, 14, and 15 as having notably high odds ratios of diseases, at >20 (Appendix A). For instance, types 5, 11, and 12 exhibited high odds ratios for cardiovascular diseases, whereas the odds ratios for IBD were markedly higher in types 14 and 15. Type 12 showed higher odds ratios for other diseases, such as malignant diseases (both under and after treatment) and hypertension compared with those of the other types. Of these five types with relatively high associations with diseases, type 5 showed greater α-diversity (both Chao1 and Shannon indices) than that of the other types, including type 10 (Appendix A). The α-diversity of types 11 and 12 based on both Chao1 and Shannon indices were also higher than those for type 10; however, α-diversity of types 14 and 15 were lower than that of type 10 (Appendix A). In regards to β-diversity, based on Bray–Curtis dissimilarity, the distance between type 10 and types 14 or 15 was significantly different; however, the distance between type 10 and types 5, 11, and 12 did not significantly differ (Appendix A). The characteristic bacterial genera for which the relative abundance was higher compared with that for other types was *Oscillospira* in type 5, *Ruminococcus* of the family *Ruminococceae* in type 11, *Streptococcus* in type 12, *Bifidobacterium* in type 14, and *Veillonella* in type 15 (Appendix A).

## 4. Discussion

In the present study, we enrolled 1803 Japanese subjects (including 15.7% healthy subjects and 85.3% with various disease statuses), and we analyzed gut microbiota using fecal samples. Characteristically, the participants with inflammatory bowel disease and functional gastrointestinal disordes, which have already been reported to show alterations in the composition of the intestinal microbiota, were included (13.1%). In addition, participants with at least one lifestyle-related disease (63.0%), such as hypertension, dyslipidemia, hyperuricemia, T2D, and obesity, were enrolled.

In this study, the gut microbiota community of these Japanese individuals was shown to stratify into enterotypes. Five possible constellations were identified based on gut microbial structure using PAM-based clustering. Importantly, the SVM-based prediction model showed fair reproducibility of these clusterings. Data at the genus level were used for community typing, and only frequently detected genera (those detected in more than 50% of the samples) were selected. As shown in Figure 3 and Table 3, the taxonomic features of the gut microbiota in Japanese subjects were characterized based on the presence of the enterotype rich in the genus *Bifidobacterium* (Type D). Another four PAM-identified communities were similar to those previously reported from Western countries [5,7]. Among the enterotypes reported by Vandeputure et al. [7], the B1 enterotype, which is the most common enterotype in healthy populations, corresponds to the PAM-based type B in the present study, which, after type E, included the second largest number of healthy individuals. Meanwhile, the B2 enterotype is similar to the PAM-based type C cluster, which was characterized by a high proportion of *Bacteroides* and a low proportion of *Faecalibacterium.* The type E community had the largest number of healthy individuals and was characterized by a rich abundance of the genus *Prevotella*, suggesting that it may be similar to the previously reported P enterotype. Finally, type A included an abundance of unclassified genera belonging to the family *Ruminococcaceae*, which is similar the R enterotype.

Nishijima et al. [12] also demonstrated that the gut microbiome structure in a Japanese population was considerably different from that of other populations, with at least one of the differences being the rich abundance of the genus *Bifidobacterium*. Park et al. [18] also described the high abundance of *Bifidobacterium* as a feature of the gut microbiota of 1596 healthy Japanese individuals. Thus, a high abundance of *Bifidobacterium* seems to be notable in the structure of Japanese gut microbiota, representign an important genus, independent of the presence or absence of various diseases.

It is well known that several *Bifidobacterium* strains are considered to be probiotic microorganisms due to their beneficial effects, and they have been included as bioactive ingredients in functional foods—mainly dairy products but also in food supplements and pharma products [19]. The beneficial effects on human health related to the consumption of *Bifidobacterium* have mainly been associated with the prevention and treatment of intestinal diseases, including IBD and IBS, as well as immunological disorders, such as allergic diseases. A recent study revealed that the administration of *Bifidobacterium bifidum* G9-1 (BBG9-1) improves quality of life for Japanese patients with constipation, as demonstrated by changes in stool consistency [20].

However, analysis regarding the association between varying disease status and types of the gut microbiota community demonstrated that a cluster rich in the genus *Bifidobacterium* is not necessarily indicative of low risk for various diseases. In the current study, type D, the cluster with a high abundance of *Bifidobacterium*, showed high odds ratios with various disease statuses, including cardiovascular diseases; hepatic diseases; IBD and functional gastrointestinal disorders; psychiatric diseases; and lifestyle-related diseases, such as hypertension, dyslipidemia, hyperuricemia, and diabetes. The functional role of *Bifidobacterium* in health status is somewhat controversial, as it has been demonstrated that it can be abundant in patients with heart failure [21], hyperlipidemia, and T2D [15,22,23]. Regardless, *Bifidobacterium*-mediated health benefits are thought to occur as a result of the complex and dynamic interplay among *Bifidobacterium*, other members of the gut microbiota, and the human host. Accordingly, our findings suggest that a gut environment that leads to the rich abundance of *Bifiodobacterium* could be harmful, rather than *Bifiodobacterium* itself exerting harmful effects.

With regards to type D communities, lactate-producing bacteria (including *Bifidobacterium*) are more abundant than other bacteria types, indicating that more lactate would be produced in this community type. Lactate produced in the colon is normally metabolized by lactate-utilizing bacteria to short-chain fatty acids (SCFAs) that are beneficial for the host, such as butyrate and propionate. However, in cases of a community with low numbers of lactate-utilizing bacteria, lactate can accumulate and exert an array of deleterious effects [24]. For example, its low pKa can drive down gut pH, which may lead to changes in the microbiota and acidosis in the colon. Lactate can also be a growth substrate for sulfate-reducing gut bacteria [25]; therefore, it has the potential to promote the formation of toxic concentrations of hydrogen sulfide. Indeed, lactate has been shown to accumulate in the colon of patients with severe colitis [26,27,28]. Type D, with the lowest α-diversity, may have fewer lactate-utilizing bacteria. Therefore, higher odds ratios for various diseases support the possibility of lactate accumulation in the gut microbiome of this type.

In the present study, we attempted to classify gut microbiota using a DMM model, as well as PAM. In the analysis based on the DMM model, increased numbers of subjects are expected to result in a larger number of communities being identified. Consistent with this expectation, we confirmed 15 clusters in the current study using a DMM model, which was a three-fold larger number of communities than that identified using the PAM-based clustering. Type 10 of the DMM model was a *Prevotella*-rich cluster similar to type E of the PAM-based clustering. Meanwhile, type 12 may be unique as a *Streptococcus*-rich cluster and had high disease odds ratios across the board. Additionally, *Streptococcus* tended to be abundant in types 14 and 15 and may be important for raising the risk of disease in these types, as *Streptococcus* is well-known to exhibit high homolactic acid fermentation and high lactic acid production. The current study has several limitations. Although we were able to enroll a larger number of subjects than that enrolled in previous studies evaluating human enterotypes, the sample size may still be inadequate. In particular, we were unable to perform an age-adjusted analysis, which would take into account intestinal microbiota alterations associated with aging. In addition, although dietary habit is supposed to be one of the most important factors for the appearance of specific enterotypes, we were not able to take into account a survey of dietary habits. Therefore, dietary factors relevant to each gut microbiota community cannot be discussed. Additionally, the association of various diseases with each of the gut microbiota communities was only evaluated based on current and past disease status. As a result, it is difficult to predict the risk/onset of various diseases using the results of the present study. Stratification of the human gut microbiome into separate clusters, such as enterotypes, provides an appealing method for the evaluation of microbial markers related to certain diseases; however, the reality is more complex. For instance, it is important to take into account the metabolic capacities of gut microbes in order to adequately discuss the relationship between gut microbiota communities and diseases. Unfortunately, profiles of metabolites, such as SCFAs, for each gut microbiota community are currently unknown. Thus, there is a need to analyze the association between microbial metabolites and community types of the gut microbiota in future studies.

## 5. Conclusions

In conclusion, we successfully demonstrated the typing of Japanese gut microbiota communities using PAM-based and DMM-model-based clustering. Unlike the enterotypes proposed for individuals from Western countries, the Japanese gut microbiota was stratified with a fair level of reproducibility into five community types using PAM-based clustering. The gut microbiota of the Japanese subjects were especially characterized by the presence of the genus *Bifidobacterium*. We also attempted to assess the association of various diseases with gut microbiota community typing. However, further investigations are needed to fully correlate the detailed microbial properties with the health status of individuals. We anticipate that precise enterotypes driven by age, gender, ethnicity, nutritional habits, and medication may be used as diagnostic markers in the near future, as well as being successfully correlated with the health status of individuals.

## Figures and Tables

**Figure 1 microorganisms-10-00664-f001:**
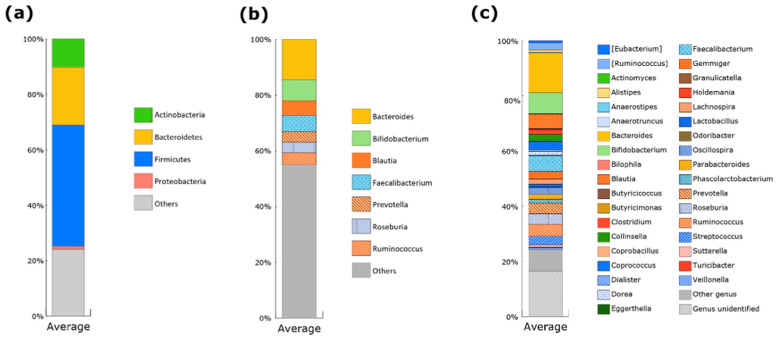
Taxonomic composition of the microbial communities of Japanese subjects enrolled in the study. Cumulative bar chart for average abundance of four major bacterial phyla (**a**), seven predominant genera (**b**), and frequently detected genera (**c**) in the gut microbiota of Japanese subjects enrolled in the study. The frequently detected genera are comprised of 36 genera, which were detected in more than 50% of the subjects.

**Figure 2 microorganisms-10-00664-f002:**
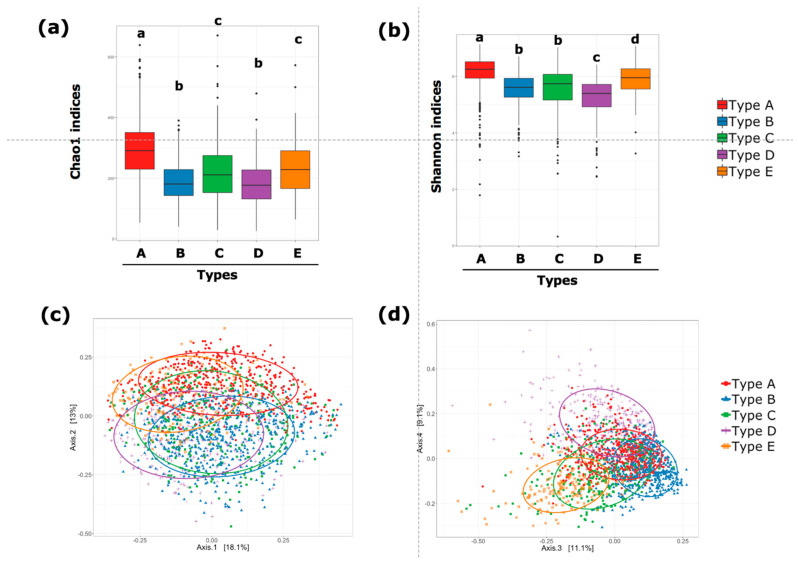
α-diversity and β-diversity of gut microbiota for five PAM-identified communities. α-diversity assessed by Chao 1 index (ASV richness estimation) (**a**) and Shannon index (ASV evenness estimation) (**b**). Statistical differences in α-diversity indices among the groups were evaluated using one-way ANOVA. Statistical significance (*p* < 0.05) is indicated by different letters. β-diversity represented by principal coordinate analysis plots based on Bray–Curtis dissimilarity. Axis 1 and axis 2 (**c**) and axis 3 and axis 4 (**d**). Ellipses enclosing the clusters indicate an 80% confidence interval. Statistically significant differences in β-diversity among the groups were confirmed using PERMANOVA (*p* = 0.001).

**Figure 3 microorganisms-10-00664-f003:**
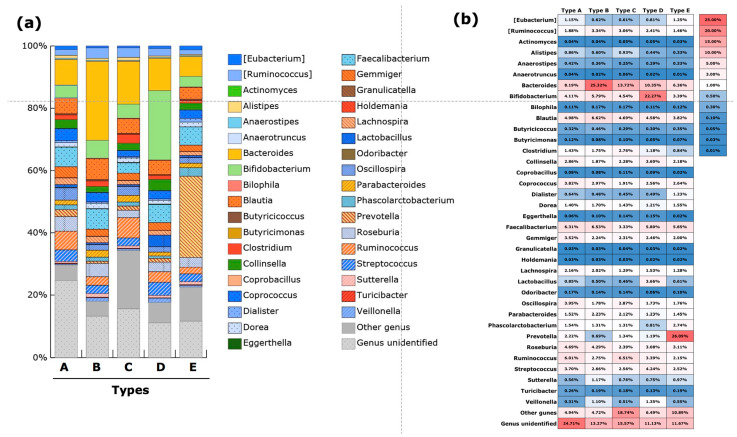
Taxonomic composition of the microbial communities at the genus level for five PAM-identified types. Cumulative bar charts of abundance of frequently detected genera (**a**) and a heatmap of the of mean abundance values of frequently detected genera (**b**). The frequently detected genera were comprise 36 genera, which were detected in more than 50% of the subjects.

**Figure 4 microorganisms-10-00664-f004:**
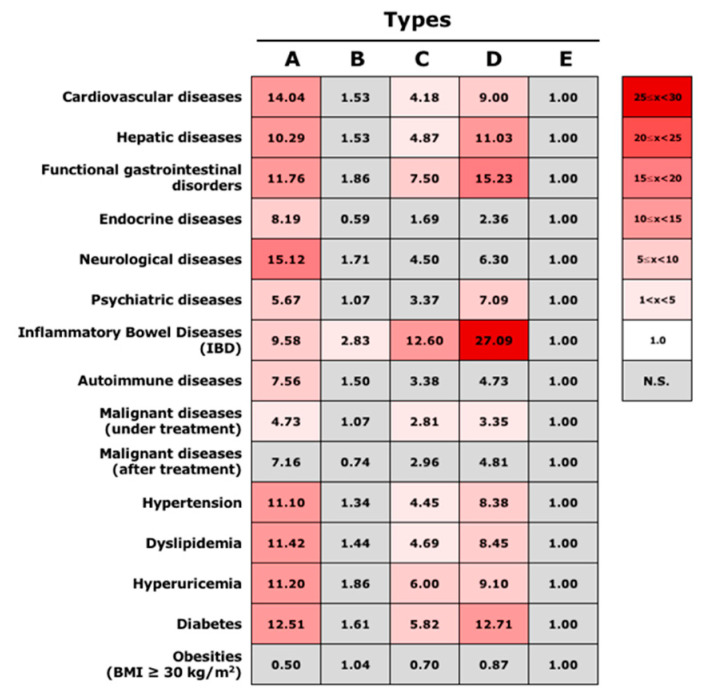
Odds ratios of various diseases in each of the PAM-identified types. The odds ratio for various diseases in each PAM-identified type was calculated based on the number of individuals with each disease. Statistically significant differences (*p* < 0.05) were calculated based on the Wald test. Red boxes indicate significantly higher odds ratios of diseases in comparison to type D. Gray boxes indicate no significant difference with type D.

**Figure 5 microorganisms-10-00664-f005:**
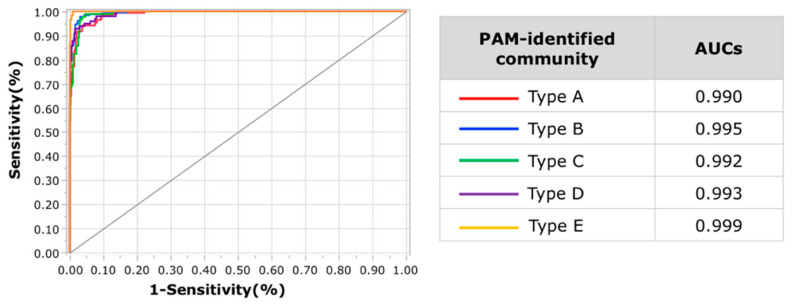
ROC curves and AUC values from the support vector machine model. Assessment of ROC curves evaluating the ability to predict each gut microbiota community type using an SVM classification model. Each curve represents the sensitivity and specificity to distinguish subjects into each community type. The area under the curve (AUC) of the ROC curve for each community type is displayed in the table on the right.

**Table 1 microorganisms-10-00664-t001:** Distribution of the enrolled study subjects.

	N	Male (Age ± SD)	Female (Age ± SD)
Total	1803	983 (63.2 ± 16.2)	820 (65.5 ± 13.4)
Healthy subjescts	283	177 (43.4 ± 11.1)	106 (49.2 ± 12.3)
Cardiovascular diseases	104	71 (74.6 ± 8.2)	33 (73.5 ± 6.9)
Hepatic diseases	168	89 (64.4 ± 12.7)	79 (69.0 ± 10.8)
Functional gastrointestinal disorders	109	61 (68.5 ± 18.0)	48 (67.8 ± 13.5)
Endocrine diseases	57	26 (68.9 ± 8.3)	31 (68.8 ± 9.1)
Neurological diseases	15	7 (66.7 ± 15.1)	8 (65.3 ± 15.5)
Psychiatric diseases	38	19 (65.5 ± 13.7)	19 (71.3 ± 13.5)
Inflammatory Bowel Diseases (IBD)	128	76 (48.4 ± 18.5)	52 (52.3 ± 15.6)
Autoimmune diseases	21	7 (72.1 ± 8.7)	14 (66.9 ± 12.3)
Malignant diseases (under treatment)	123	81 (69.2 ± 9.6)	42 (69.7 ± 8.8)
Malignant diseases (after treatment)	160	99 (71.1 ± 9.4)	61 (68.4 ± 9.5)
Hypertension	619	313 (70.2 ± 9.8)	306 (70.2 ± 9.1)
Dyslipidemia	819	422 (68.3 ± 11.4)	397 (69.1 ± 9.9)
Hyperuricemia	138	99 (68.5 ± 12.4)	39 (72.5 ± 7.9)
Diabetes	474	268 (67.4 ± 11.3)	206 (66.3 ± 10.7)
Obesities (BMI ≥ 30 kg/m^2^)	96	40 (51.2 ± 17.2)	56 (55.2 ± 15.3)

BMI; body mass index.

**Table 2 microorganisms-10-00664-t002:** Number and rate of healthy subjects in each PAM-identified type.

	The Number of Enrolled Subjects	The Number of Healthy Subjects	The Rate of Healthy Subjects (%)	Male (Age ± SD)	Female (Age ± SD)
Type A	512	25	4.9	264 (69.8 ± 13.0)	248 (69.9 ± 9.7)
Type B	552	147	26.6	299 (58.4 ± 17.1)	253 (62.9 ± 14.6)
Type C	271	28	10.3	151 (64.4 ± 16.4)	120 (66.6 ± 11.9)
Type D	292	20	6.8	133 (65.5 ± 14.9)	159 (62.4 ± 14.5)
Type E	176	63	35.8	136 (57.3 ± 15.7)	40 (62.4 ± 16.7)

**Table 3 microorganisms-10-00664-t003:** Summary of taxonomic features characteristically rich in gut microbiota of each PAM-identified type.

	Characteristic Feature	Other Features
Type A	family *Ruminococcaceae*	genera *Coprococcus*, *Gemminger* and *Roseburia*
Type B	genus *Bacteroides*	genera *Blautia* and *Faecalibacterium*
Type C	genus *Bacteroides*	genera *Megamonus*, *Fusobacterium* and *Proteus*
Type D	genus *Bifidobacterium*	genera *Lactobacillus* and *Streptococcus*
Type E	genus *Prevotella*	

## Data Availability

This study used sequence data from multiple independent studies. The data were deposited to the Sequence Read Archive (SRA) in the NCBI database with accession numbers PRJNA766337, PRJNA804422, PRJNA809418, and PRJNA809527. The latter two are available as of 1 May 2022.

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
