# Peer review of "Typing of the Gut Microbiota Community in Japanese Subjects"

_microorganisms, 2022, doi:10.3390/microorganisms10030664_

Round 1

Reviewer 1 Report

This is an interesting study on gut micorbiota and its association on various diseases. In my opinion the study is well executed, interesting and provides significant contribution to the literature. The limitation of the study are that authors have not considered age of the participants and their dietary habits and we know that gut microbiota is changed with aging and diet, however authors do recognize this in their study limitation section . Some comments to consider to improve the clarity /educational value: 

1. Line 42-43- Bacteroides are both, type B and type C? This is confusing

2, Line 40- changes in gut microbiota have been the cause and consequence of NAFLD and this should be mentioned, particularly given its association with infection as well as pandemic of fatty liver disease  ( https://www.ncbi.nlm.nih.gov/pmc/articles/PMC8087474/)

Line 83- exclusion criteria within the past 3 months- how this cut off was determined? 

Author Response

Dear the editors of Microorganisms,

We thank the editors and the reviewers for his/her thorough review of our manuscript (microorganisms-1636200). According to your reviewer’s suggestions, we have carefully revised our manuscript. We revised our manuscript using red colored text and underline. Following is a point-by-point response to the comments. We hope that revised manuscript in now suitable for publication of “Microorganisms.

To reviewer 1

Thank you for your comments. As you kindly suggested, our manuscript was revised as follows.

Comment 1. Line 42-43- Bacteroides are both, type B and type C? This is confusing.

(Response) Thank you for your important comments. Although the line pointed by Reviewer 1 may be different, we understood it is a comment on the description in the section of abstract. As you pointed out, main genus is Bacteroidetes is characteristic bacterial genus both in Type B and Type C. Of course, other bacterial genus compositions except for Bacteroidetes are quite different between Type B and Type C, and Type B corresponds to B1 type and Type C corresponds to B2 type of previously reported enterotypes (cited in main text as [7, 8]). As you commented, because the abundance of genus Bacteroides is high both in Type B and Type C, it is indeed confusing. Therefore, we summarized the detailed differences of five enterotypes in Table 4. Briefly, characteristic genera in Type B and Type C are follows; genera Bacteroides, Blautia, and Faecalibacterium (Type B), Bacteroides, Fusobacterium, and Proteus (Type C). Due to limitation of word counts, we are not able to explain these detailed differences in abstract. If you and editor allow exceeding word limits of abstract, we will add more detailed description regarding the difference between Type B and Type C. If we misunderstood your suggestion, please point it out to us.

Comment 2, Line 40- changes in gut microbiota have been the cause and consequence of NAFLD and this should be mentioned, particularly given its association with infection as well as pandemic of fatty liver disease ( https://www.ncbi.nlm.nih.gov/pmc/articles/PMC8087474/).

(Response) Thank you for your valuable comments. According to your suggestion, we revised the description and citated literatures in the section of Introduction as follows; (lines 37-41) Recently accumulated information regarding the human gut microbiota has revealed that its composition is linked to host health and various diseases, including inflammatory bowel disease (IBD), irritable bowel syndrome (IBS), and allergies, as well as lifestyle-related diseases such as obesity and nonalcoholic fatty liver disease (NAFLD) [1-4].

Line 83- exclusion criteria within the past 3 months- how this cut off was determined? 

(Response) Thank you for your constructive comments. As we totally agree with your comments, we revised the description in the section of Materials and Methods as follows; (lines 87-88) within the past 3 months prior to collecting fecal samples

We believe that the manuscript has improved after incorporation of the changes suggested by the reviewers. We hope that the revised manuscript is suitable for publication, and would look forward to the final decision.

Yours sincerely,

Tomohisa Takagi, MD, PhD.

Associate Professor

Molecular Gastroenterology and Hepatology, Graduate School of Medical Science

Kyoto Prefectural University of Medicine

Kawaramachi-Hirokoji, Kamigyo-ku, Kyoto 602-8566

Japan

Tel: + 81-75-251-5508

Fax: +81-75-251-0710

Reviewer 2 Report

First, I would like to congratulate the authors of this article. I think they have done a great job. The authors have spent a lot of time and effort.

In reference to the use of English, the language in general is correct, however, I think it could use a review by a native English speaker. This would help improve certain parts and give a more natural touch to the use of English.

Minor concerns:

Introduction: What about?: Hosoda, S., Nishijima, S., Fukunaga, T. et al. Revealing the microbial assemblage structure in the human gut microbiome using latent Dirichlet allocation. Microbiome 8, 95 (2020). https://doi.org/10.1186/s40168-020-00864-3.

The discussion is considered rather poor in relation to the findings of the present study. There is no discussion at all about the peculiarities of the studied individuals (Table 1). It would be also, more attractive to readers if the discussion includes a summary of possible mechanisms for the appearance of specific enterotypes.

References: Please review format and fix it.

Author Response

Dear the editors of Microorganisms,

We thank the editors and the reviewers for his/her thorough review of our manuscript (microorganisms-1636200). According to your reviewer’s suggestions, we have carefully revised our manuscript. We revised our manuscript using red colored text and underline. Following is a point-by-point response to the comments. We hope that revised manuscript in now suitable for publication of “Microorganisms.

To reviewer 2

Thank you for your comments. As you kindly suggested, our manuscript was revised as follows.

Comment 1

Introduction: What about?: Hosoda, S., Nishijima, S., Fukunaga, T. et al. Revealing the microbial assemblage structure in the human gut microbiome using latent Dirichlet allocation. Microbiome 8, 95 (2020). https://doi.org/10.1186/s40168-020-00864-3.

(Response) Thank you for your constructive comments. According to your comments, we revised the description and cited the above manuscript in the section of Introduction as follows;

(lines 65-67) Accordingly, the stratification of gut microbiota on the dataset used in Nishijima et al. [12] revealed that the distribution of enterotypes was also characteristic in Japanese compared to other countries [13].   

Comment 2

The discussion is considered rather poor in relation to the findings of the present study. There is no discussion at all about the peculiarities of the studied individuals (Table 1). It would be also, more attractive to readers if the discussion includes a summary of possible mechanisms for the appearance of specific enterotypes.

(Response) Thank you for your valuable comments. According to your suggestion, we added the sentences regarding the peculiarities of the enrolled individuals in the section of Discussion as follows;

(lines 350-358) In the present study, we enrolled 1,803 Japanese subjects (included 15.7% healthy subjects and 85.3%various diseases statuses) and we analyzed the gut microbiota using fecal samples. Characteristically, the participants with inflammatory bowel disease and functional gastrointestinal disordes, which have already been reported to show the alterations in the composition of the intestinal microbiota, were included (13.1%). In addition, the participants with at least one lifestyle-related disease (63.0%) such as hypertension, dyslipidemia, hyperuricemia, T2D, and obesity were enrolled.

In this study, the gut microbiota community of these Japanese individuals was shown to stratify into enterotypes.

Furthermore, we agree with the importance of possible mechanisms for the appearance of specific enterotypes. However, unfortunately, it is so far very hard to describe the mechanisms due to the limitations written in the discussion part, especially lack of the data for dietary habit of subjects in the present study. Therefore, we added this issue in the limitation section according to your comments as below.   

(lines 435-436) In addition, although dietary habit is supposed to be one of the most important factors for the appearance of specific enterotypes, we were not able to take into account a survey on dietary habits.

Comment 3

References: Please review format and fix it. 

(Response) Thank you for your constructive comments. As we totally agree with your comments, we revised the format in the section of References.

We believe that the manuscript has improved after incorporation of the changes suggested by the reviewers. We hope that the revised manuscript is suitable for publication, and would look forward to the final decision.

Yours sincerely,

Tomohisa Takagi, MD, PhD.

Associate Professor

Molecular Gastroenterology and Hepatology, Graduate School of Medical Science

Kyoto Prefectural University of Medicine

Kawaramachi-Hirokoji, Kamigyo-ku, Kyoto 602-8566

Japan

Tel: + 81-75-251-5508

Fax: +81-75-251-0710
